# Integrated Graphene Oxide with Noble Metal Nanoparticles to Develop High-Sensitivity Fiber Optic Particle Plasmon Resonance (FOPPR) Biosensor for Biomolecules Determination

**DOI:** 10.3390/nano11030635

**Published:** 2021-03-04

**Authors:** Chien-Hsing Chen, Chang-Yue Chiang, Chin-Wei Wu, Chien-Tsung Wang, Lai-Kwan Chau

**Affiliations:** 1Department of Biomechatronics Engineering, National Pingtung University of Science and Technology, Pingtung 91201, Taiwan; garychc@mail.npust.edu.tw; 2Graduate School of Engineering Science and Technology and Bachelor Program in Interdisciplinary Studies, National Yunlin University of Science and Technology, Yunlin 64002, Taiwan; 3Bachelor Program in Interdisciplinary Studies, National Yunlin University of Science and Technology, Yunlin 64002, Taiwan; wucw@yuntech.edu.tw; 4Department of Chemical and Materials Engineering, National Yunlin University of Science and Technology, Yunlin 64002, Taiwan; 5Department of Chemistry and Biochemistry and Center for Nano Bio-Detection, National Chung Cheng University, Chiayi 62102, Taiwan

**Keywords:** fiber optic, particle plasmon resonance, gold nanoparticles, graphene oxide, anti-IgG, IgG, biosensor

## Abstract

In this research, a direct, simple and ultrasensitive fiber optic particle plasmon resonance (FOPPR) biosensing platform for immunoglobulin G (IgG) detection was developed using a gold nanoparticle/graphene oxide (AuNP/GO) composite as signal amplification element. To obtain the best analytical performance of the sensor, experimental parameters including the surface concentration of GO on the AuNPs, formation time of the GO, the concentration of the anti-IgG and incubation time of anti-IgG were optimized. The calibration plots displayed a good linear relationship between the sensor response (ΔI/I_0_) and the logarithm of the analyte concentrations over a linear range from 1.0 × 10^−10^ to 1.0 × 10^−6^ g/mL of IgG under the optimum conditions. A limit of detection (LOD) of 0.038 ng/mL for IgG was calculated from the standard calibration curve. The plot has a linear relationship (correlation coefficient, R = 0.9990). The analytical performance of present work’s biosensor was better than that of our previously reported mixed self-assembled monolayer of 11-mercaptoundecanoic acid/6-mercapto-1-hexanol (MUA/MCH = 1:4) method by about three orders of magnitude. The achieved good sensitivity may be attributed to the synergistic effect between GO and AuNPs in this study. In addition, GO could immobilize more antibodies due to the abundant carboxylic groups on its surface. Furthermore, we also demonstrated that the results from this sensor have good reproducibility, with coefficients of variation (CVs) < 8% for IgG. Therefore, the present strategy provides a novel and convenient method for chemical and biochemical quantification and determination.

## 1. Introduction

Clinical disease detection requires testing tools for real-time diagnosis and precision medicine, which are helpful in the early detection of disease, effective prevention and treatment, thereby improving the success rate and mortality of patients in clinical monitoring and treatment [1]. Therefore, it is urgent to develop a highly sensitive, highly selective, simple and rapid detection method to determine the existence of disease biomarkers in clinical samples (e.g., blood and biological tissues). Optical sensors have high sensitivity, excellent non-specificity and multifunctionality as well as real-time monitoring of the affinity among biomolecules. Therefore, various optical sensors have been developed in recent years and used in medical detection, including the surface plasmon resonance (SPR) technique for real-time monitoring of heart medicine on cardiac muscle cell function and behavior [2], surface-enhanced Raman scattering (SERS) plus a microfluidic system for detecting prostate-specific antigen (PSA) [3], fluorescence immunoassay (FIA) for detecting biomarkers (adenosine) in the urine of lung cancer patients [4], colorimetric assays for detecting heart disease bioindicator troponin I (cardiac troponin I, cTnI) [5], fluorescent semiconductor quantum dot (QD) for detecting the influenza virus (influenza H1N1 virus) [6], radioimmunoassay (RIA) for detecting insulin in plasma [7] and enzyme-linked immunosorbent assay (ELISA) for detecting carcinoembryonic antigen (CEA) [8]. However, these optical detection techniques have some disadvantages such as high prices, time requirements, the need for specific operators, unstable isotopes, large sizes and difficult field assays.

Responding to this challenge, a new-generation optical sensor technology was developed by combining a fiber optic assembly with noble metal nanoparticles, which is known as fiber optic particle plasmon resonance (FOPPR) sensor technology [9]. The principle is to use the evanescent wave generated in the total reflection transfer of light in the optical fiber core to excite Au nanoparticles in the sensing region (the fiber core surface), so that the free electrons on the Au nanoparticle surface can perform the particle plasmon resonance (PPR) phenomenon of collective dipole oscillation, also known as the localized surface plasmon resonance (LSPR) phenomenon. The PPR phenomenon will change the resonance energy as the environmental refractive index or dielectric constant changes, making it suitable for testing and analyzing the interaction of molecules among different molecule species. Prior studies have indicated that the FOPPR sensor can be successfully used in various applications of chemical and biochemical domains and that it is characterized by real-time detection, high specificity and high speed. It has been experimentally proven that the sensitivity of this technology can reach the nanomolar (nM) level in bioassays [10,11,12,13,14,15,16]. The method has many advantages; however, in order to face the detection standard for precision medicine in clinical detection, how to enhance the sensitivity and reproducibility are current topics. At present, sensing platforms have developed according to the sandwich hybridization detection method and competition method to enhance their sensitivity [14,15,16], but the fabrication procedure is complicated, and the stability requires further enhancement.

Therefore, this study proposed a new concept of self-assembled monolayers and used the direct detection method to develop the sensing chip. The method consisted of quantitative testing using FOPPR sensor technology based on a Au nanoparticle (AuNP)/graphene oxide (GO) composite as the signal amplifying element for detecting immunoglobulin G (IgG). The GO was a plane lamina of a 2D hybrid material [17], which was a composite structure consisting of sp^2^ and sp^3^ hybridized carbon atoms and defects that had oxygen-containing functional groups such as –O–, –OH and –COOH. The GO had particular physical and chemical characteristics [18] such as high biocompatibility, a large specific surface area and abundant π conjugation, which is convenient for surface modification, thus making it easy for the GO to immobilize biomolecules. It has been extensively used in biosensors [19,20,21,22]. In 2010, Wang et al. used a graphene SPR sensor to test biomolecules. The result showed that the gold thin film surface-modified graphene sensing chip had higher sensitivity than the conventional gold foil sensing chip. The increase in sensitivity was attributed to the increased adsorption of biomolecules on the graphene and the optical properties of graphene [23].

In this study, we demonstrated a simple, rapid and versatile in situ approach for the fabrication of graphene oxide by using a modification of the Hummers’ method, which does not involve the use of surfactants [24,25]. Finally, under optimal experimental conditions (Figure 1), a wide linear range, high reproducibility and a low limit of detection (LOD) were presented, and the obtained LOD was much lower than the level reported in prior references. In addition, compared with the existing mixed self-assembled monolayer of 11-mercaptoundecanoic acid (MUA)/mercaptohexanol (MCH), the bridging-based molecule detection of IgG effectively increased the detection sensitivity by three orders of magnitude, mainly because of the synergetic effect between AuNPs and GO in the study [26]. Additionally, due to the abundant carboxyl groups on the surface, the GO could immobilize more antibodies. This study is the first one to establish FOPPR based on using a AuNP/GO composite as a signal amplifying element to detect IgG. This study provides a novel cost-effective and convenient method for chemical and biochemical quantitative testing.

## 2. Materials and Methods

### 2.1. Materials and Chemicals

All of the chemicals were used as received and did not require further purification. Cetyltrimethylammonium bromide (CTAB), N-hydroxy-succinimide (NHS), 1-ethyl-3-(3-dimethylaminopropyl)-carbodiimine hydrochloride (EDC), hydrogen tetrachloroaurate (III)tetrahydrate (HAuCl_4_·4H_2_O), 11-mercaptoundecanoic acid (MUA; ≥95%), 6-mercapto-1-hexanol (MCH; ≥97%), (3-mercaptopropyl)-trimethoxysilane (MPTMS, 98%), graphite flakes (99% carbon basis), hydrochloric acid, anti-IgG and IgG (Isoelectric point (pI) = 7) were bought from Sigma-Aldrich (St. Louis, MO, USA). Ethanol and acetone were bought from Hy Biocare Chem (New York, NY, USA). Cystamine dihydrochloride was purchased from Acros (Geel, Belgium). Sodium citrate was bought from J.T. Baker (Phillipsburg, NJ, USA). Hydrogen peroxide (H_2_O_2_) and sulfuric acid (H_2_SO_4_; ≥98%) were bought from Fluka (Buchs, Switzerland). Cystamine dihydrochloride (C_4_H_12_N_2_S_2_∙2HCl) was bought from Acros (Geel, Belgium). Potassium permanganate (KMnO_4_; ≥99%) was obtained from Alfa Aesar (Tewksbury, MA, USA). During the whole experimental process, all of the water solutions were prepared using ultrapure water from the Milli-Q purification system (with a resistivity of 18.2 MΩ cm, Millipore, Bedford, MA, USA). All biological samples were configured in phosphate buffered saline (PBS) buffer (pH = 7.4). Multimode plastic-clad silica optical fiber with respective core and cladding diameters of 400 and 430 μm was purchased from Newport (model F-MBC, Irvine, CA, USA). The optical fiber probe and sensing chip (poly (methyl methacrylate) (PMMA) plates) were prepared using a CO_2_ laser engraving machine (New Taipei, Taiwan).

### 2.2. Preparation of GO

The GO in this study was prepared by synthesizing graphite flakes using the improved Hummers method [24,25]. The experimental process is described below. First, 1 g graphite flakes, 84 g H_2_SO_4_ and 6 g KMnO_4_ were mixed in a three-necked round flask (in a 0 °C ice bath). Afterwards, the heating temperature was set at 35 °C, and the heating time was 2.5 h. The mixture was mixed with 40 g ultrapure water slowly in batches, and the temperature was increased to 90 °C and heated for 30 min. Finally, the mixture was mixed with 8 g H_2_O_2_ (30%) and reacted at 45 °C for 10 min (the solution turned from dark brown to light yellow). After the product had cooled to room temperature, it was washed with a 1-M HCl water solution and then washed with ionized water until the solution became neutral. The product was dried in a freezing vacuum for 8 h. The morphological and structural characterizations of the GO were observed using a JEOL JSM-7610F Plus field-emission scanning electron microscope (FESEM, Tokyo, Japan).

### 2.3. Preparation and Characterization of the AuNP Probe

AuNPs were synthesized according to previously reported procedures [27]. The characteristic peak was determined by a Jasco V-570 UV–Vis–NIR spectrophotometer (Tokyo, Japan) and verified using JEM-2100Plus transmission electron microscopy (TEM, Tokyo, Japan) images, through which the actual shape and size of the nanoparticles were measured. Afterwards, as stated above, the AuNPs were self-assembled on the uncovered part of the optical fiber [28], an image of the AuNPs on the optical fiber was taken by a field-emission scanning electron microscope and the size distribution of the AuNPs was determined according to the SEM image. The microchannel chip preparation and packaging processes were performed according to prior work [29]; it was cleaned with deionized (DI) water, dried with nitrogen and stored at room temperature.

### 2.4. Preparation of the AuNP–GO–Anti-IgG Probe

The preparation of the AuNP–GO–anti-IgG probe is shown in the following Figure 1. First of all, a 0.02-M cystamine solution was prepared and poured into the microfluidic chip for a four-hour reaction. Afterward, it was cleaned with DI water, and the GO was modified on the surface of the AuNPs. In order to obtain the optimal sensor analysis performance, the experimental parameters were discussed, including the surface concentrations (0.01%, 0.05%, 0.1%, 0.15% and 0.2%) of GO on the AuNPs and the immobilization time of the GO (1, 2, 4, 6 and 8 h). Afterwards, a mixed solution of EDC (0.2 M) and NHS (0.05 M) was prepared and poured into the microfluidic chip for 30 min of activation (the –COOH group and EDC/NHS), and the GO was modified to activate the functional group. The anti-IgG concentrations (10, 20, 50, 80 and 100 μg/mL) and anti-IgG incubation times (1, 2, 4, 6 and 12 h) were adjusted, after which the solution reacted with a 1 M ethanolamine (EA) water solution at a pH of 8.5 for 10 min and the unreacted sites were inactivated. Finally, the PBS solution was poured in for preservation.

### 2.5. Microchannel Chip and FOPPR Sensing System

The FOPPR sensing system structure is shown in Figure 2, including (a) a light emitting diode (LED) driver circuit (self-made); (b) an LED (model IF-E93, Industrial Fiber Optic, Inc., wavelength 530 nm); (c) a microfluidic chip (the sensor system requires only 30 µL of sample for detection in a microfluidic channel); (d) a photodiode (S1336-18BK, Hamamatsu); (e) a photoreceiver amplification circuit (PAC, home-made); (f) a data acquisition card lock-in module (dynamic signal acquisition module USB-9234 and Lab-VIEW software, National Instrument) and (g) a computer. The LED driver circuit generated drive signals, provided a fixed 1K frequency and fixed voltage to drive the LED and provided a reference signal for the photoreceiver amplification circuit. The light signal coupling was transferred through the optical fiber optical waveguide into the microfluidic chip, and the light energy transferred in the sensing region was influenced by the LSPR absorption band of the AuNPs, resulting in changes in the light signals. The photodiode received light at the optical fiber end, and the light signal captured by the photodiode was processed by the photoreceiver amplification circuit and transferred to the lock-in amplifier for operations. Finally, the computer read out the response values.

The working principle is to use the evanescent wave generated in the total internal reflection (TIR) of light in the optical fiber core to excite Au nanoparticles on the fiber core surface to induce particle plasmon resonance (PPR) of AuNPs. When light propagates in the fiber core by virtue of multiple TIRs, the evanescent wave on the fiber core surface excites the PPRs of immobilized AuNPs, and thus, the light transmitted through the fiber is attenuated. Hence, the FOPPR biosensing platform for real-time monitoring of molecular interactions is based on the change in localized evanescent wave absorption by the AuNPs upon molecular binding, resulting in decreased transmission intensity measured at the distal end of the optical fiber. Here, our sensor response is defined as (I_0_ − I_S_)/I_0_ = 1 − I_S_/I_0_ = ΔI/I_0_, where the collected optical signal of a sensor immersed in an analyte solution (I_S_) is compared to the intensity of the sensor immersed in a blank solution (I_0_), and I_S_/I_0_ is analogous to transmittance.

Our previous study has shown that the FOPPR sensing platform can be operated under ambient conditions without temperature control. There is a negligible influence of the ambient temperature fluctuation (1 °C merely causes a variation in ΔI/I_0_ of about 0.0025) on the sensitivity of the sensor [14].

### 2.6. Sample Preparation

Stock standard solution of IgG with a concentration of 1.0 × 10^−5^ g/mL was prepared in PBS at pH 7.4 and stored in a freezer at −20 °C until use. By diluting the stock solution, the concentration range of the standard IgG solution was changed from 1.0 × 10^−10^ to 1.0 × 10^−6^ g/mL and then stored at 4 °C for future use. The IgG was injected into the sensing chip from low concentration to high concentration during detection and the response was monitored and recorded instantly. The result of three replications of all data was the average ± standard deviation (SD), and MATLAB 2020b (MathWorks) and Origin 2020b (OriginLab) were used for statistical analysis. The signal response and concentration were drawn to obtain the calibration curve.

## 3. Results and Discussion

### 3.1. Preparation, Surface Analysis and Characteristics of the Sensing Probe

In order to confirm the successful synthesis of AuNPs and GO material and the preparation of the AuNP/GO-functionalized sensing probe, various instruments were used for verification, including a UV–Vis–NIR spectrophotometer, image verification using transmission electron microscopy (TEM), field-emission scanning electron microscopy (FESEM) and EDS.

The absorption spectrum of the AuNP solution was tested using a UV–Vis–NIR spectrophotometer, as shown in Figure 3a. The absorbance of the AuNP solution was adjusted to 1 absorbance unit (a.u.), and the maximum peak was at 521.03 ± 0.8 nm. The verification by TEM is shown in Figure 3b, wherein the AuNPs are complete balls. There was no aggregation of AuNPs. As shown in Figure 3c, the mean particle size of the AuNPs was about 12.00 ± 0.78 nm, and the standard deviation was about 0.83 nm. Additionally, the structures of the synthetic GO, AuNP and AuNP–Cys–GO were verified by FESEM. Figure 3d shows a thin, wrinkly, paper-like structure, which is typically observed in graphene oxide-based materials. Figure 3e shows the FESEM image of the AuNPs. It could be observed that the AuNPs were mostly spherical and narrowly distributed over the optical fiber surface. The mean diameter of the AuNPs was 16.88 ± 2.8 nm (with at least 100 particles per batch, Figure 3f). Figure 3g shows the FESEM image of the modified AuNPs–GO on the optical fiber. It was obvious that the AuNPs were covered with thin, gray GO and that the coexistence of an AuNP/GO thin film was formed.

In this study, we used UV–Vis spectroscopy to monitor the molecular modification process [13,30,31,32,33,34]. In order to know the characteristic peak generated in the absorption spectrum of the modified AuNPs–GO on the optical fiber, a self-made fiber optic spectrometer was used for verification. The spectrometer (Ocean Optics, HR4000, optical resolution: 0.025 nm) was excited by a white light source (Ocean Optics, LS-1, 6.5 W) for measurement, as shown in Figure 4a. Each step of the functionalization was validated by the fiber optic spectrometer measurement result, including the gradual functionalization of AuNPs, AuNP–cystamine, AuNP–Cys–GO, AuNP–Cys–GO–anti-IgG and AuNP–Cys–GO–anti-IgG–IgG. The simple AuNPs showed a characteristic peak and wavelength at 523 nm. The AuNP–cystamine, AuNP–Cys–GO, AuNP–Cys–GO–anti-IgG and AuNP–Cys–GO–anti-IgG–IgG showed characteristic peaks at about 523, 530, 532 and 533 nm, respectively. A shift in peak wavelength (λ^max^) and an increase in peak extinction coefficient (Ext^max^) were observed for each functionalization step, which is consistent with an increase in the local refractive index [30,31,35]. These red shifts and increases in Ext^max^ results proved the change in the local refractive index near the AuNP surface, thereby indirectly proving the steps of immobilization of antibodies on the AuNP surface or the antibody–analyte interaction. In addition, the absorbance peak was increased. Prior reports have observed increased absorption and spectral red shift after the modification of chemical and biological molecules on the AuNP surface [13,36]. These results demonstrate that a chemisorbed monolayer on a AuNP–GO-modified sensing probe with an appropriate receptor can be used to transduce ligand–receptor binding at a surface into an extinction change with a sensitivity that is useful for biosensor applications. Therefore, the results of this study proved the successful functionalization of the AuNP–GO-modified sensing probe. Additionally, the surface chemical composition of the AuNPs–GO was analyzed by EDS. As shown in Figure 4b, the surface chemical constituents included C, O and Au, meaning that the GO was successfully fixed to the surface of the AuNPs.

### 3.2. Optimization of Analytical Conditions

In order to obtain the optimal sensor analysis performance, the optimal experimental parameters were discussed, including the surface concentration of GO on the AuNPs, the GO formation time, the anti-IgG concentration and the anti-IgG incubation time. The first step of each experiment was to inject a PBS buffer solution as the experimental initial baseline (I_0_). The IgG concentration was fixed at 1.0 × 10^−7^ g/mL, and the sensor signal response was △I. The experimental result is shown in Figure 5. When the GO concentration was 0.1%, the signal response (△I/I_0_) reached its maximum (Figure 5a). The signal response decreased as the GO concentration increased because excessive GO covered the surface of the AuNPs. The AuNPs required higher energy for generating the PPR effect; therefore, 0.1% was selected as the optimal GO concentration. In addition, the GO formation time was discussed. The effects of the GO formation time on the sensing response were checked at immobilization times of 1, 2, 4, 6 and 8 h. The sensor response intensity (△I/I_0_) increased gradually with the GO immobilization time and became stable after four hours, which resulted from obvious steric hindrance, and the antibody activity declined (Figure 5b). Therefore, a four-hour GO immobilization time was used in subsequent work. The carboxyl group of the GO was activated by EDC/NHS. When the GO surface was activated, anti-IgG at different concentrations was injected into the microfluidic chip so as to select the optimal experimental conditions. The covalence between the amine and the carboxyl group was connected by chemical bonding to fix the antibody to the GO. The effect of the antibody concentration on the sensor was measured under different concentrations (10, 20, 50, 80 and 100 μg/mL). It can be observed in Figure 5c that the signal response increased with the anti-IgG concentration and then stabilized after 50 μg/mL, meaning the abundant carboxyl group of GO met the ability of antibody fixation. Therefore, an anti-IgG concentration of 50 μg/mL was used in subsequent work. Figure 5d shows the anti-IgG incubation time (1, 2, 4, 6 and 12 h). The signal intensity increased gradually with the incubation time and became stable after 4 h (Figure 5d). Therefore, four hours was selected as the incubation time for determining the anti-IgG. Generally speaking, the optimal biochip modification conditions of IgG were tested by the aforesaid experimental optimization. The concentration of GO was 0.1%, the immobilization time was 4 h, the anti-IgG modification concentration was 50 μg/mL and the modification time was 4 h.

### 3.3. Non-Specific Adsorption

Non-specific adsorption is the most important issue in the development of biosensors. In order to prevent non-specificity from combining and blocking the residual activated groups on the AuNP–GO probe, the ethanolamine passed through the sensor surface. Our previous study [10] indicated this method to be effective. The specificity of the biological sensing chip was measured by comparing with the detection of bovine serum albumin (BSA), cardiac troponin I (cTnI) and IgG. As shown in Figure 6, compared with the background signal, the FOPPR sensor intensity variation of BSA (1.0 × 10^−6^ g/mL) and cTnI (1.0 × 10^−6^ g/mL) was almost negligible, and the injected IgG (1.0 × 10^−10^ g/mL) apparently presented the dynamic curve generated by anti-IgG and IgG bonding, which was attributed to the specific bonding of anti-IgG and IgG.

Therefore, besides using anti-non-specific adsorption layers on both the surfaces of the fiber core and AuNPs, the buffer pH and pI values had to be selected to avoid electrostatic interaction. In this study, the adsorption of monoclonal mouse IgG in a PBS buffer system at pH 7.4 seems to be governed mainly by electrostatic interactions, as the retained IgG is neutrality charged with pI values mostly around 7 [37,38,39,40].

### 3.4. Sensitivity of the AuNPs–GO–Anti-IgG Probe to IgG Detection

After the AuNPs–GO–anti-IgG modification procedure was optimized, quantitative testing of the IgG was performed. First, PBS was injected as a baseline; then, the IgG at different concentrations was injected for the test. According to the real-time monitoring graphs shown in Figure 7a, the signal response increased with the IgG solution concentration increase, meaning that the FOPPR sensor based on AuNPs/GO/anti-IgG could quantize the ability of IgG at different concentrations. Under the optimum conditions, in the linear range of 1.0 × 10^−10^ to 1.0 × 10^−6^ g/mL IgG, the calibration curve displayed a good linear relationship between the sensor response (ΔI/I_0_) and the log of the analyte concentration. According to the standard correction curve shown in Figure 7b, the LOD of the IgG was 0.038 ng/mL and the correlation coefficient R = 0.9990. The analysis performance of the AuNPs–GO–anti-IgG probe was increased by 22.8 times compared with the performance of the mixed self-assembled monolayer (SAM) (MUA/MCH = 1:4) anti-IgG probe (taking an IgG concentration of 1.0 × 10^−6^ g/mL as an example) (Figure 7c). The detected concentration range was enhanced by three orders of magnitude (Figure 7d). The inferior performance of our previous method using a mixed SAM (MUA/MCH) was most likely a result of the steric hindrance and an unstable mixed self-assembled monolayer. Additionally, there were abundant carboxyl groups on the surface of the GO, which could fix more antibody molecules.

Finally, the previously reported method [41] was used to estimate the molecular binding kinetic analysis and determine the antigen–antibody affinity and binding kinetic constant. Four kinds of IgG at different concentrations were used to obtain the kinetic association rate constant (ka) and dissociation rate constant (kd) by linear fitting, which were 8.98 × 10^5^ M^−1^s^−1^ and 1.36 × 10^−2^ s^−1^, respectively. Afterwards, the ka and kd values were used to calculate the affinity constant K_f_ (wherein K_f_ = ka / kd) and (6.58 ± 0.38) ×10^7^ M^−1^ (n = 3). The result showed that this study was close to the SPR-calculated antigen–antibody affinity and kinetic binding constant as well as the measured SPR value [42], proving that the AuNP–GO probe developed in this study could assess real-time biomolecular binding interaction.

### 3.5. AuNP–GO Repeatability and Reproducibility Analysis

In order to study the repeatability of the AuNP–GO probe, the sensing chip based on AuNPs–GO–anti-IgG was evaluated. The measurement was repeated three times under optimum conditions. The experimental results showed that the method had high reproducibility, and that in the concentration range of 1.0 × 10^−10^ to 1.0 × 10^−7^ g/mL of IgG, the coefficient of variation (CV) was lower than 7.79%. Additionally, the preservation stability of the AuNPs–GO–anti-IgG probe was evaluated. The sensing chip based on AuNPs/GO–anti-IgG was stored in 4 °C PBS (pH = 7.4) for two weeks and then injected with 1.0 × 10^−7^ g/mL IgG to determine the IgG signal responses at different storage times, as shown in the following Figure 8. The result showed that the IgG signal response remained at 96.84% within seven days, and the signal was still 92.1% after two weeks, proving that the probe had good long-term stability within two weeks. To sum up, the aforementioned result proves that the proposed AuNPs–GO–anti-IgG probe has good stability and repeatability that allows the preparation of the sensing probes without a tight schedule. For long-term storage purposes, numerous studies in the literature have reported convenient and widely used methods for long-term storage of proteins [38,43,44,45,46,47]. These preservation methods will be explored in our future sensor prototype development to improve preservation time and stability.

Many promising results have already been demonstrated, but we believe that there is still room for improvement. The chemical aspects for improvement include the gold nanoparticle size, the density of gold nanoparticles on the probe surface, optimization of the receptor density on the gold nanoparticles’ surface and structure of the functionalized monolayer on the Au nanoparticle surface, the surface concentration of GO on the AuNPs and the formation time of GO on the AuNPs. In addition, fundamental understanding of the underlying biochemistry, surface chemistry, physics chemistry and material chemistry and technological advances are needed in order to enhance the sensor performance and to improve the reliability, stability and functionality of GO-based FOPPR biosensors in real applications. Moreover, we should focus on validation of assay reliability on complex real samples. We believe that the system can be widely applied to other clinically or environmentally important biological molecules.

## 4. Conclusions

This study is the first to present a FOPPR sensing platform using a probe based on a AuNP/GO composite. The quantitative result of the IgG test showed an LOD of 0.038 ng/mL and the correlation coefficient R = 0.9990. Furthermore, we also demonstrated that the results from this sensor have good reproducibility, with coefficients of variation (CVs) < 8% for IgG. It not only provided a qualitative result for rapid detection but also provided a quantitative result for determining the IgG in the samples. To sum up, the AuNP–GO probe provided an accurate and highly sensitive detection method that could be used in clinical applications of precision medicine in the future to help doctors diagnose and monitor patient conditions as well as implement early detection and early treatment.

## Figures and Tables

**Figure 1 nanomaterials-11-00635-f001:**
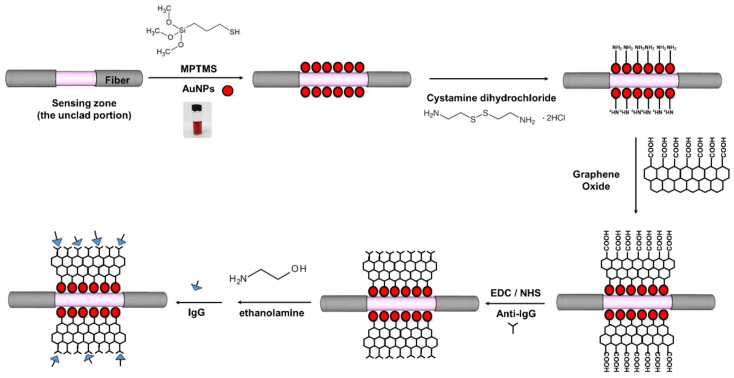
Schematic of the steps involved in the fabrication of sensor fiber.

**Figure 2 nanomaterials-11-00635-f002:**
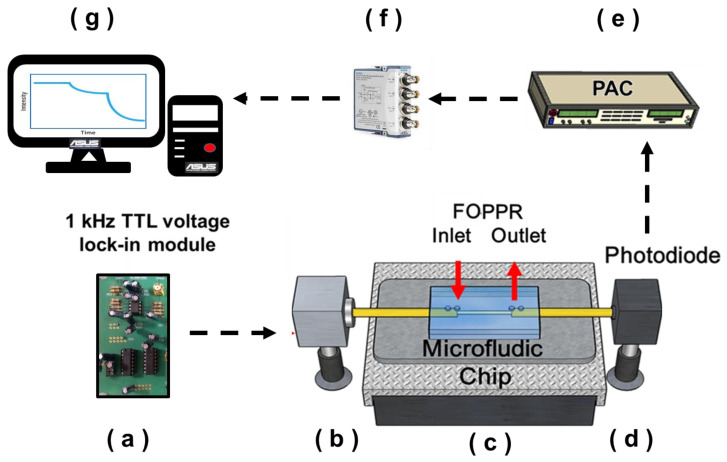
Schematic representation of the experimental setup used for the fiber optic particle plasmon resonance (FOPPR) biosensor. The setup consists of (**a**) a light emitting diode (LED) driver circuit; (**b**) a light emitting diode; (**c**) a microfluidic chip; (**d**) a photodiode; (**e**) a photoreceiver amplification circuit; (**f**) a data acquisition card and (**g**) a computer.

**Figure 3 nanomaterials-11-00635-f003:**
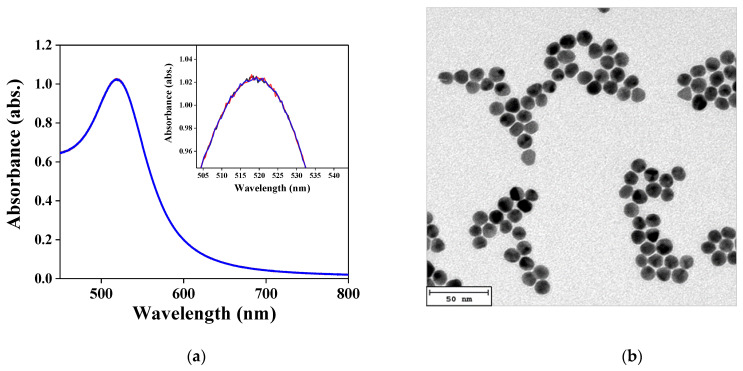
Structural characterizations of materials. (**a**) Absorption spectra of Au nanoparticles (AuNPs) in aqueous medium in the visible region; (**b**) TEM image of AuNPs; (**c**) size distribution of AuNPs by TEM image analysis; (**d**) SEM image of graphene oxide (GO); (**e**) SEM image of AuNPs on the fiber core surface; (**f**) size distribution of AuNPs by SEM image analysis; (**g**) SEM image of AuNPs/GO on the fiber core surface.

**Figure 4 nanomaterials-11-00635-f004:**
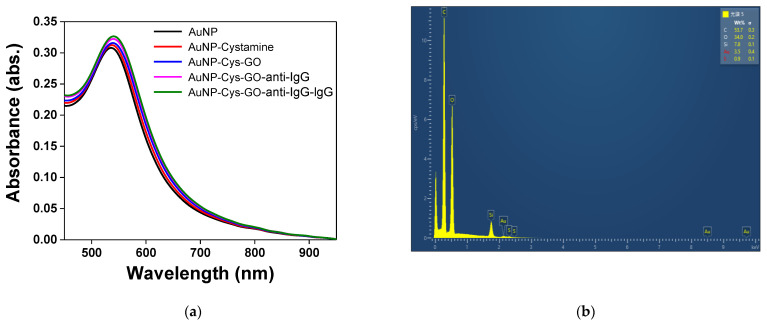
(**a**) Gradual functionalization of AuNP, AuNP–cystamine, AuNP–Cys–GO, AuNP–Cys–GO–anti-IgG and AuNP–Cys–GO–anti-IgG–IgG in the probe modification spectrogram; (**b**) EDS measurement verification.

**Figure 5 nanomaterials-11-00635-f005:**
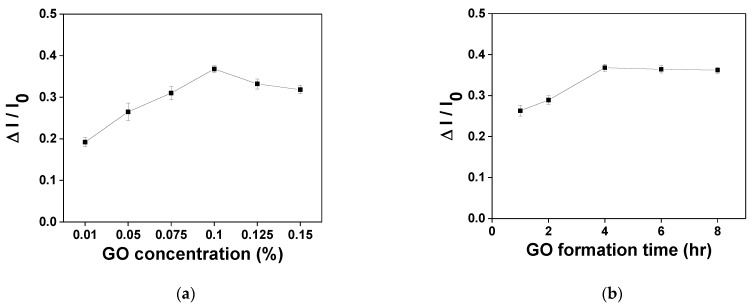
(**a**) Effect of GO concentration ratio in the starting solution on the sensor response. (**b**) Effect of immersion time given to form GO on the sensor response. (**c**) Effect of concentration of anti-IgG used for the bioconjugation process on the sensor response. (**d**) Effect of incubation time of anti-IgG used for the bioconjugation process on the sensor response (n = 3).

**Figure 6 nanomaterials-11-00635-f006:**
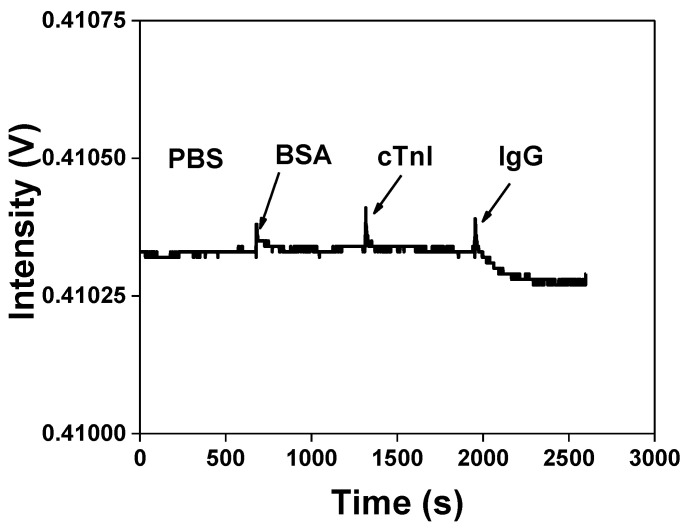
Non-specific adsorption tests. The sensorgram of IgG antibody-functionalized AuNPs–GO FOPPR sensor in response to bovine serum albumin (BSA, 1.0 × 10^−6^ g/mL), cardiac troponin I (cTnI, 1.0 × 10^−6^ g/mL) and IgG (1.0 × 10^−10^ g/mL) solutions.

**Figure 7 nanomaterials-11-00635-f007:**
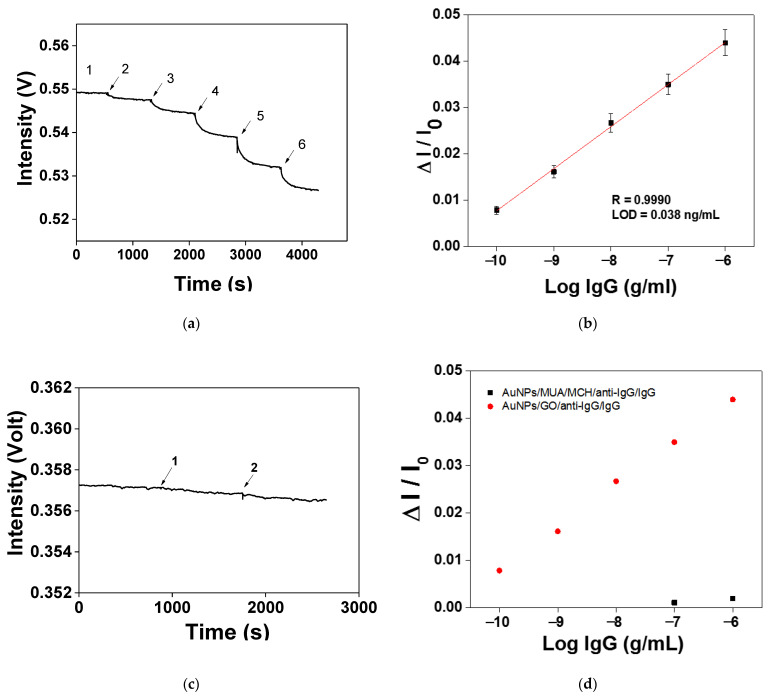
(**a**) Temporal response of the anti-IgG functionalized AuNP–GO probe signal with serial injection of standard solutions with different immunoglobulin G (IgG) concentrations of (1) 1.0 × 10^−10^, (2) 1.0 × 10^−9^, (3) 1.0 × 10^−8^, (4) 1.0 × 10^−7^ and (5) 1.0 × 10^−6^ g/mL. (**b**) Calibration curve for IgG by anti-IgG-functionalized AuNP–GO probe. (**c**) Temporal responses of the anti-IgG-functionalized AuNPs–MUA/MCH probe with serial injection of standard solutions with different IgG concentrations of (1) 1.0 × 10^−7^ and (2) 1.0 × 10^−6^ g/mL. (**d**) Comparison between anti-IgG-functionalized AuNP–GO probe and the anti-IgG-functionalized AuNPs–MUA/MCH probe in the FOPPR system.

**Figure 8 nanomaterials-11-00635-f008:**
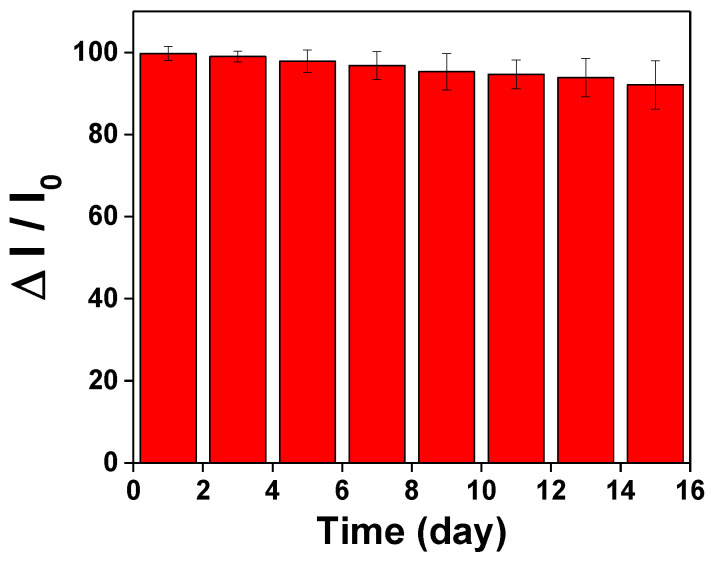
Storage stability of the AuNP–GO–anti-IgG probe onto sensing chip.

## Data Availability

Not applicable.

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
