# Peer review of "Integrated Graphene Oxide with Noble Metal Nanoparticles to Develop High-Sensitivity Fiber Optic Particle Plasmon Resonance (FOPPR) Biosensor for Biomolecules Determination"

_nanomaterials, 2021, doi:10.3390/nano11030635_

Round 1

Reviewer 1 Report

Article „Integrated graphene oxide with noble metal nanoparticles to develop high sensitivity fiber optic particle plasmon resonance (FOPPR) biosensor for biomolecules determination” deals with obtaining Au/GO composite coating an optical fiber. To GO bonded anti-IgG and then detected IgG.

In general, this article has the potential to be published after some corrections.

 The confirmation of molecular bonding should be done by FTIR and not be based on small shifts of a broad UV-Vis peak.

“The simple AuNPs 229 showed a characteristic peak and wavelength at 523 nm. The AuNP-cystamine, AuNP- 230 Cys-GO, AuNP-Cys-GO-anti-IgG and AuNP-Cys-GO-anti-IgG-IgG showed 231 characteristic peaks at about 523 nm, 530 nm, 532 nm and 533 nm, respectively” Please provide the resolution of this spectrometer during the time of measurements. DO not forget that this line is very broad. Highly unlikely that a change of 1 nm could be confirmed and not belonging to the system and not to thermal instability. Two separate measurements gave you results (Fig 3a) 521 ± 0.8 nm and (Fig 4a) 523 nm for AuNP . For the bonding studies, FTIR should be applied.

The description of chapter 2.5 should be done more thoroughly, especially the description of what was measured exactly - intensity at a specific point at maximum. Did the device accommodate for changes in pH temperature over time, which could shift the absorption peak?  

Multiple times the word „method” is used

„In this research process, a simple and rapid method for preparing GO was demon-96 strated. The method improved on the Hummers method for preparation [24,25], and the 97 method was free of surfactants.”

The misleading sentence “It could be observed that the AuNPs were spherical and uniformly 213 and compactly distributed over the optical fiber surface”. There are also triangle and trapezoidal-shaped particles. Fig 3b shows it exactly.

Fig 3e  twice e letter used, repace with g

Fig 4b EDX measurement does not sum up to 100%. What was removed from the data?

“96.84% within seven days, and the signal was still 92.1% after two weeks, proving the 342 probe had good long-term stability within two weeks.” “AuNPs-GO-anti-IgG probe had good stability and 344 repeatability. “ It reaches 0% after 6.6 months. If it is good, then compare it with other sensors on the market.

32 references are not enough to claim thoroughly done literature research. People are working in this field for years. Above that, you have at least 14 self-citations on 32 references 43.7% (not acceptable).

Author Response

Dear Reviewer:

We would like to thank you for the thoughtful critiques of our manuscript. We have taken your comments fully into account and revised the manuscript accordingly. Our point-by-point responses to your comments are appended to this letter, beginning on the next page. List of changes made in the revised manuscript based on your comments. In the following, your comments are in black Arial font while our responses are in blue Times Roman font. We have also highlighted the changes made in the revised manuscript (with changes in red Times Roman font). Thank you very much for your kind consideration of this paper.

Reviewer 2 Report

In the manuscript by Chien-Hsing Chen et al., the authors present research on integrated graphene oxide with noble metal nanoparticles to develop high sensitivity fiber optic particle plasmon resonance biosensor for biomolecules determination. In my opinion, the paper is publishable in Nanomaterials. The authors should address the following issues before this work can be accepted for publication.

  1. Page 1, line 27. LOD?
  2. Page 1, line 27. 0.038 ng/mL? See, Fig. 7(b) and legend - 0.042 ng/mL.
  3. Page 1, line 27. ml or mL? See all the manuscript.
  4. Page 1, line 28. r=0.9981? See, Fig. 7(b) and legend – R2=0.9981 – determination coefficient.
  5. Page 1, line 30. MUA/MCH?
  6. Page 4, line 154. D.I. ---> DI – see line 150.
  7. Page 4, lines 158 and 160. ECD? ECD ---> EDC.
  8. Page 5, Fig. 2. FO-PPR ---> FOPPR.
  9. Page 5, line 194. Citation Examples for OriginLab Products https://www.originlab.com/index.aspx?go=Company&pid=1130. See also Matlab. https://www.researchgate.net/post/how_to_cite_Matlab. https://www.mathworks.com/matlabcentral/answers/414438-how-do-i-cite-matlab-in-a-bibliography-or-a-published-journal.
  10. Page 7, Fig. 3. (e) ---> (g)
  11. Page 7, Fig. 3. (c) and (f) are similar. Counts or Number? Size or Particle diameter. Lowercase or uppercase? Legend or no legend. Please standardize!!!!!!!!!!!!
  12. Page 9, Fig. 5. In all plots on y-axis is DeltaI/I0. Wouldn't the following dependencies (GO formation time) vs DeltaI/I0, anti-IgG vs. DeltaI/I0, and (anti-IgG incubation time) vs. DeltaI/I0 be better?
  13. Page 9, Fig. 5. [anti-IgG] ---> anti-IgG concentration. Or [anti-IgG] ---> anti-IgG (see Fig. 7(b) and (d)).
  14. Page 9, Fig. 5. I suggest using the same scale for all plots in the case of DeltaI/I0, i.e. 0.00-0.05. See Fig. 7(b) and (d).
  15. Page 10, Fig. 6. I would narrow down the scale of y-axis. For example 0.4100-0.4105?
  16. Page 10, line 306. 0.038 ng/mL? see above – point 2.
  17. Page 10, line 309. r=0.9981? see above – point 3.
  18. Page 11, Fig. 7, the second row. (a) ---> (c). (b) ---> (d).
  19. Page 11, Fig. 7(c), title of x-axis. (sec) ---> (s). See Fig. 7(a).
  20. Page 11, Fig. 7(d), scale of y-axis, DeltaI/I0. Correct scale from 0.00 upto 0.05.
  21. Page 12, Fig. 8, x-axis. day ---> Time (d).
  22. Page 12, line 363. 0.038 ng/mL? see above – point 2.
  23. Page 12, line 364. r=0.9981? see above – point 2.

Author Response

(The authors gave the same response as above.)

Round 2

Reviewer 1 Report

Text corrections and the authors' answers pleased me fine.

Reviewer 2 Report

The authors have made a substantial improvement for this article. The manuscript can be accepted for publishment in the present form.